# Effects of high-intensity respiratory muscle training on respiratory muscle strength in individuals with Parkinson's disease: Protocol of a randomized clinical trial

**Sherindan Ayessa Ferreira de Brito**[☯], **Aline Alvim Scianni**[☯], **Bruna Mara Franco Silveira**[☯], **Elem Rodrigues Martins de Oliveira**[☯], **Maria Eduarda Mateus**[☯], **Christina Danielli Coelho de Morais Faria**[iD][☯]*

Department of Physical Therapy, Universidade Federal de Minas Gerais, Belo Horizonte, Minas Gerais, Brazil

☯ These authors contributed equally to this work.
* cdcmf@ufmg.br

**Data Availability Statement:** No datasets were generated or analysed during the current study. All

## Abstract

### Objective

To investigate the efficacy of high-intensity respiratory muscle training (combined inspiratory and expiratory muscle training) in improving inspiratory and expiratory muscle strength, inspiratory muscle endurance, peak cough flow, dyspnea, fatigue, exercise capacity, and quality of life in this population.

### Methods

A randomized controlled trial, concealed allocation, blinded assessments, and intention-to-treat analysis will be carried out. Altogether, 34 individuals with PD (age $\geq$ 50 years old, with maximum inspiratory pressure (MIP) <80cmH$_2$O or maximum expiratory pressure (MEP) <90cmH$_2$O) will be recruited. Patients will be randomly assigned to either (1) high-intensity respiratory muscle training (experimental group, 60% of MIP and MEP) or (2) sham training (control group, 0cmH$_2$O). Individuals will perform a home-based intervention, with indirect home supervision, consisting of two daily 20-min sessions (morning and afternoon), seven times a week, during eight weeks. Primary outcomes are MIP and MEP. Secondary outcomes are inspiratory muscle endurance, peak cough flow, dyspnea, fatigue, exercise capacity, and quality of life. The effects of the training will be analyzed from the collected data using intention-to-treat. Between-group differences will be measured using a two-way ANOVA with repeated measures (2*3), considering baseline, post-intervention, and 12-week follow-up.

### Impact

The results of this trial will provide valuable new information on the efficacy of high-intensity respiratory muscle training in improving muscle strength, functional outcomes, and quality of life in individuals with PD. Performing combined inspiratory and expiratory muscle training

relevant data from this study will be made available upon study completion.

**Funding:** Our research was funded by Coordenação de Aperfeiçoamento de Pessoal de Nível Superior (CAPES), Fundação de Amparo à Pesquisa do Estado de Minas Gerais (FAPEMIG), Conselho Nacional de Desenvolvimento Científico e Tecnológico (CNPq) and Pró-reitoria de Pesquisa da Universidade Federal de Minas Gerais (PRPq/UFMG). All authors were fully involved in the study and preparation of the manuscript. Each of the authors has read and agreed with the content of the final manuscript and approved its submission. The material within has not been and will not be submitted for publication elsewhere in whole or in part in any language, except as an abstract. Therefore, the work is not under review elsewhere and has not been previously published.

**Competing interests:** The authors have declared that no competing interests exist.

using a single equipment is cheaper and feasible, takes less time and is easy to use. In addition, this intervention will be carried out in the home environment that increases accessibility, reduces time, and costs of transport, which increases the feasibility to reproduce their findings in clinical practice.

## Trial registration

NCT05608941. Registered on November 8, 2022.

## Introduction

Parkinson's disease (PD) is the second most prevalent neurodegenerative disease worldwide [1]. Individuals with PD commonly have a significant reduction in respiratory muscle strength [2, 3] and inspiratory muscle endurance [3], and it can intensify with the disease progression [2]. Reduced respiratory muscle strength is associated with an impaired cough efficacy [4]. Cough is the main mechanism of protection of the airways, a process that occurs through a high-velocity airflow. This airflow is generated by the contraction of the expiratory muscles while the glottis is closed [4, 5]. Expiratory muscle weakness results in reduced subglottic pressure during the compression phase, decreased expiratory airflow, and inadequate clearance of material from the airway [4, 5]. This favors the aspiration of foreign bodies and accumulation of mucus, and consequently, the occurrence of respiratory infections, such as pneumonia [5]. Individuals with PD have a significant reduction of cough efficacy when compared to controls [5]. This is especially important for these individuals since pneumonia is the leading cause of death in this population [6].

Decreased respiratory muscle strength and endurance may be also associated with dyspnea perception on exertion (minimum or average) and orthostatic dyspnea [7]. Dyspnea is an underestimated symptom in individuals with PD [8]. Studies suggested that around 40% these individuals have dyspnea [8, 9]. Dyspnea is multifactorial and arises from the interaction of several factors, such as the effects of the disease on the respiratory and cardiovascular systems, sensory and perceptual changes, and other associated comorbidities [7]. This may result from a mismatch between central respiratory motor activity and income afferent information from receptors in the airways, chest wall structures, and respiratory muscles [7, 10]. The inability of the respiratory muscles to meet the increased respiratory demand, as occurs during activity performance, can cause dyspnea [7]. Finally, respiratory muscle weakness has already been associated with the occurrence of fatigue, reduced exercise capacity and quality of life in this population [2, 11].

Respiratory muscle training has been used to increase inspiratory and expiratory muscle strength in people with PD [12, 13]. It was found only three randomized controlled trials that have investigated the effects of respiratory muscle training in individuals with PD [14–16]. These studies showed low methodological quality when evaluated by the PEDro scale [12, 14–16]. To our knowledge, only one study investigated the effects of inspiratory muscle training for the following outcomes (inspiratory muscle endurance, dyspnea and quality of life). However, this study did not show the effect size and the data provided did not allow its calculation [12, 14, 17]. The effect size provides information about the magnitude of the effects of an intervention, being an estimate of the size of the difference between groups [18]. Interventions that have larger effect sizes are more likely to produce significant outcomes for patients. Therefore, it is essential to determine the effect size of the interventions that will be used in clinical practice [18]. Finally, although respiratory muscle weakness is associated with the fatigue and

reduced exercise capacity in individuals with PD [2, 11], the effects of respiratory muscle training on these outcomes have not yet been investigated.

Respiratory muscles respond to training stimuli in a similar fashion to other skeletal muscles, so when their fibers are overloaded, both the ratio of Type I and the size of Type II fibers are increased [19]. Previous studies that investigated the effects of inspiratory muscle training in individuals with PD, showed increases in MIP, ranging from $-1.28cmH_2O$ [14] to $-16cmH_2O$ [16]. In addition, increases in MEP from $15.5cmH_2O$ [16] to $27.97cmH_2O$ [15] was observed when expiratory muscle training was performed. In a study that have investigated the effect of high-intensity respiratory muscle training in individuals after stroke, an improvement in MIP of $-43cmH_2O$ and MEP of $51cmH_2O$ was found, that is, approximately a double [20]. Similar results were observed in other populations when training of higher intensities were applied, such as heart failure and chronic obstructive pulmonary disease [21, 22]. In addition, the benefits of respiratory muscle training on other functional outcomes (perception of dyspnea, fatigue, and exercise capacity) seem to be obtained when a high training volume (load, duration, and frequency) is applied [21, 22]. However, the effects of respiratory muscle training of higher intensities (load, intensity, and volume) still need to be investigated in individuals with PD.

Finally, respiratory muscle training is task-specific, so the increase in inspiratory and expiratory pressure depends on training the inspiratory and expiratory muscles, respectively. Combined inspiratory and expiratory muscle training seems to be more effective, since with elevated breathing, both muscle groups are increasingly recruited [23]. In addition, initiation of a cough usually results in a large inspiration followed by expulsive events, in which the expiratory muscles contract. Therefore, so for an effective cough, the coordination between inspiratory and expiratory muscles is necessary [4]. Therefore, it is important to investigate the effects of including both types of exercises to obtain the full benefits of training [12]. In addition, performing combined inspiratory and expiratory muscle training using a single equipment is cheaper and feasible, takes less time and is easy to use. To the best of our knowledge, no study has investigated the effects of combined inspiratory and expiratory muscle training.

Thus, the primary aim of this study will be to investigate the effects of high-intensity respiratory muscle training (combined inspiratory and expiratory muscle training) on inspiratory and expiratory muscle strength in individuals with PD. The secondary aim of this study will be to investigate the efficacy of high-intensity respiratory muscle training (combined inspiratory and expiratory muscle training) in improving inspiratory muscle endurance, peak cough flow, dyspnea, fatigue, exercise capacity, and quality of life in this population.

## Materials and methods

### Design

A prospective, superiority parallel-group randomized controlled trial, with concealed allocation, allocation ration 1:1, blinded assessments, and intention-to-treat analysis will be carried out in a community-based setting in the city of Belo Horizonte/Brazil. Community-dwelling individuals with PD will be recruited and randomly assigned to receive either (1) high-intensity respiratory muscle training (experimental group) or (2) sham training (control group). Fig 1 shows the schedule of enrolment, interventions, and assessments [24].

All subjects will be informed of the study procedures and will provide written consent. A trained examiner will collect the outcome measures at baseline (week-0), post-intervention (after the 8-week intervention), and one month after the cessation of the intervention (12-week follow-up) (Fig 2) [24, 25]. Data collection and analysis will be carried-out by a researcher, blinded to the group allocation. All individuals will receive an identification code

| | STUDY PERIOD | | | | |
|---|---|---|---|---|---|
| | Enrolment | Allocation | Post-allocation | | Close-out |
| TIMEPOINT** | $-t_1$ | 0 | $t_1$ | $t_2$ | $t_x$ |
| ENROLMENT: | | | | | |
| Eligibility screen | X | | | | |
| Informed consent | X | | | | |
| Allocation | | X | | | |
| INTERVENTIONS: | | | | | |
| [Experimental group] | | | ●────────● | | |
| [Control group] | | | ●────────● | | |
| ASSESSMENTS: | | | | | |
| [Sociodemographic data] | X | | | | |
| [Primary outcomes variables] | X | | | X | X |
| [Secondary outcomes variables] | X | | | X | X |

**Fig 1. Schedule of enrolment, interventions, and assessments.**

to ensure anonymity. The study obtained ethical approval from the Institutional Research Ethical Committee (CAAE: 53970421.0.0000.5149; number: 5.356.570) of the [information was included in the unmasked version]. The trial was prospectively registered at the ClinicalTrials. gov (NCT05608941).

## Participants

A non-probabilistic sample will be recruited from the community through contact with health centers, research groups and university extension programs. Individuals will be included according to the following criteria: age ≥50 years; idiopathic PD diagnosed by a neurologist; taking anti parkinsonian medication, and who have been medically stable for at least six months; classified between stages 1–3 of the modified Hoehn & Yahr Scale; ability to walk independently, with or without assistive devices [26]; MIP <80cmH$_2$O or MEP <90cmH$_2$O (suggested values to eliminate significant muscle weakness) [27].

**Elegibility confirmed**
Informed consent obtained

**Baseline assessment
(Week-0)**
*Primary outcomes:* MIP and MEP
*Secondary outcomes:* Inspiratory
muscle endurance, peak cough flow,
dyspnea, fatigue, exercise capacity
and quality of life

**Randomization**
(34 individuals)

**Experimental group**
High-intensity respiratory muscle
strength (60% da MIP and MEP)
20 minutes per session
2 x per day
7x per week
8 weeks

**Control group**
Sham respiratory muscle strength
($0 cmH_2O$)
20 minutes per session
2 x per day
7x per week
8 weeks

**Post-intervention assessment
(week-8)**
*Primary outcomes:* MIP and MEP
*Secondary outcomes:* Inspiratory
muscle endurance, peak cough flow,
dyspnea, fatigue, exercise capacity
and quality of life

**Follow-up assessment (week-12)**
*Primary outcomes:* MIP and MEP
*Secondary outcomes:* Inspiratory
muscle endurance, peak cough flow,
dyspnea, fatigue, exercise capacity
and quality of life

**Fig 2. Flow diagram of the planned protocol.** MIP: Maximum Inspiratory Pressure; MEP: Maximum Expiratory
Pressure.

Individuals will be excluded according to the following criteria: if they were engaged in a respiratory muscle training protocol in the last four weeks; possible cognitive impairment as determined by cutoff scores (in points) of the Mini-Mental Status Examination according to education level reference [28] use deep brain stimulation (DBS); smokers or who stopped smoking less than six months ago; have been affected by respiratory or cardiac infections in the last month; had any other neurological, musculoskeletal, cardiovascular or respiratory disorders that could affect their ability to perform the tests [14–16, 20].

Physical therapists, who will monitor both interventions (experimental and control), will be eligible to work with the participants if they have at least two years of professional experience in the area of neurological or respiratory rehabilitation. In addition, these physical therapists will be trained by experts in this type of training.

## Participant withdrawal

Participants may withdraw from the study for any reason at any time. Researchers can withdraw participants from the trial for safety reasons or not conducting training. A maximum of one week of non-performance will be allowed [25].

## Randomization

A trained research assistant will generate the randomization sequence prior to the commencement of the study. Individuals will be randomly allocated through blinded, computer-generated randomization in six-participant blocks. Group assignment will be concealed in sequentially numbered and sealed opaque envelopes. After the baseline measures, eligible participants will be randomly allocated (experimental or control groups), when the contents of the envelopes will be revealed by the trained treating therapist [25].

## Intervention

Participants will undergo high-intensity inspiratory and expiratory muscle training (experimental group), or a sham training, which will consist of the same protocol, but without load (control group). The training load will be the only difference between the two groups. Individuals will receive a home-based intervention, split into two daily 20-min sessions (morning and afternoon), totaling 40 min per day, seven times a week, during eight weeks. Each daily session will be composed into four blocks of three minutes, with a two-minute rest between blocks. The Orygen Dual Valve® (Forumed S.L., Girona, Spain) device will be used to carry out the intervention, which provides inspiratory workloads from $0cmH_2O$ to $70cmH_2O$ and expiratory workloads from $0cmH_2O$ to $80cmH_2O$.

Participants in the experimental group will perform a high-intensity inspiratory and expiratory muscle training (60% of MIP and MEP). The initial training load for each participant will be set at 60% of his/her maximal baseline MIP and MEP for both inspiratory and expiratory strength training, respectively. Once a week, the trained researcher will visit their homes to perform the necessary procedures to guarantee the intensity of the training. On this visit, MIP and MEP will be evaluated and the training load will be progressed to ensure that 60% of the new pressure values are maintained [16, 29]. Borg score of dyspnea and effort [30] will be considered for adjusting training intensity, and scores from 4 to 6 will be targeted. Borg score will be asked after one set of three minutes, and if Borg score is <4 or >6 the training load will be adjusted to higher or lower than 60% of MIP and MEP [31]. The participants will be trained and instructed to perform the exercise program on their own with no supervision. In order to maintain the participants blinded to the training load, the device will be covered with opaque material.

In the control group, a sham intervention will be implemented: the initial resistance of the device will be 0cmH$_2$O, and will be maintained throughout the intervention period—there will be no load progression. All procedures adopted with experimental group, including the weekly home visit, will also be performed with individuals in the control group. However, there will be no real change in the training load. All devices will be wrapped with an opaque material so that the load or possible respiratory training load is not visualized.

The protocol will be home-based intervention, meaning it will not be directly supervised. To monitor adherence to the protocol, individuals will receive a training diary, in which duration and subjective perception of effort will be recorded using the Borg scale [30], for each intervention day. A caregiver will be instructed to assist the individual in completing the diary, if necessary. In baseline, all the participants will be evaluated and receive all the information regarding the home-based training in a research laboratory.

## Procedures

Demographic, anthropometric, and clinical information will be obtained, followed by the collection of the primary and secondary outcomes. These data will be collected by trained researcher, who will be blinded to the group allocation. Outcome measurement will be carried out in the university laboratory.

### Primary outcomes measures

The primary outcomes to be measured are inspiratory and expiratory muscle strength. Maximum inspiratory and expiratory pressures (MIP and MEP, respectively) will be measured using a digital manovacuometer (LEB-LabCare/UFMG, Brazil) [32, 33]. To measure the MIP, individuals will be instructed to perform inspiratory maneuvers from residual volume to total lung capacity [34]. A minimum of five measurements will be performed, with a one-minute rest between them. A difference of less than 10% must be achieved among the three best measurements and the last measurement of the test cannot be the greatest [34].

To measure the MEP, individuals will be instructed to perform inspiratory maneuvers from total lung capacity to residual volume [34]. The number of measurements and the reproducibility criterion will be the same as previously described for inspiratory muscle strength [34].

The MIP and MEP values will be recorded and reported in cmH$_2$O. The measurement will be carried out following the recommended guideline for the use of the manovacuometer [34].

### Secondary outcomes measures

The secondary outcomes to be measured are inspiratory muscle endurance, peak cough flow, dyspnea, fatigue, exercise capacity and quality of life. Inspiratory muscle endurance will be measured using a flow-resistive loading device (POWERbreathe® KH2). Individuals will be asked to breathe against a sub-maximal inspiratory load (50% of the maximal inspiratory pressure evaluated in baseline) until task failure [31]. Tests lasting from 2 to 7 minutes will be accepted. Tests that do not meet this criterion will be repeated in a different day with adjustments on resistance. Endurance test duration, number of breathes, total work, pressure, flow, power, volume, and tension time index will be recorded and used in the analyzes [31]. The measurement will be carried out following the recommended guideline for the flow-resistive loading tests [34].

Peak cough flow measurement will be performed with the peak expiratory flow meter (Mini-Wright Peak Expiratory Flow Meter) [34–37]. The individual will be asked to cough as vigorously as possible [36]. At least three measures will be taken, which must have a maximum difference of 5% between them. The highest value, in L/min, will be recorded and used in the

analysis [35, 36]. The measurement will be carried out following the recommended guideline for the flow-resistive loading tests [34].

Dyspnea will be measured using the instrument of the Medical Research Council (MRC) [38]. The participants will be asked to choose which level best represents how dyspnea limits their activities of daily living, in a 5-point scale, in which 0 indicates 'breathless only with strenuous exercise' and 4 indicates 'too breathless to leave the house'. This measurement tool has adequate measurement properties for assessing dyspnea in this population [38]. The MRC score will be used in the analyses.

Fatigue will be measured using the Fatigue Severity Scale (FSS) [39, 40]. The FSS is composed of nine statements, and participants will be asked to rate how much they agree with each of the statements. For each item, the scores range from 1 to 7, where 1 strongly disagrees and 7 strongly agrees. A higher score indicates a higher level of fatigue. The FSS is recommended by the Movement Disorders Society to assess fatigue in individuals with PD [39] and has adequate measurement properties for assessing this outcome in this population [39, 40]. The FSS score will be used in the analyses.

Exercise capacity will be measured using the Six-minute Walk Test (6MWT). Participants will be instructed to walk as far as possible in a 30-meter hallway for six minutes [41]. The 6MWT has adequate measurement properties for assessing exercise capacity in this population [42]. The test will be carried out using a standardized protocol for 6MWT [41]. Two tests will be performed on the same day, with a 30-min interval between them. The test with the greatest distance (in meters) walked will be considered and used in the analyses [41].

Quality of life will be measured using the Parkinson's Disease Questionnaire-39 (PDQ-39) [43]. The PDQ-39 is composed of 39 items divided into eight dimensions, and for each item the score ranges from 0 to 4 [43]. The score for each dimension varies from 0 to 100. The PDQ-39 has adequate measurement properties for assessing exercise capacity in this population [43].

## Data monitoring

An independent researcher, who will be blind to the group allocation, will be responsible for monitoring any adverse effects, database management, and statistical analyses. Participants will be asked to report any adverse effects or discomfort, which will be registered and reported [25]. Physical therapists will be responsible for the monitoring of exercise intensity and participants program compliance.

## Sample size

The sample size calculation was performed considering the primary outcome measures (inspiratory and expiratory muscle strength). The sample size was estimated using the G*Power Software® (version3.1) and considering data provided by previous randomized controlled trials that investigated the effects of respiratory muscle training on respiratory muscle strength in individuals with PD [14, 15].

The effect size ($f$) of 1.00 for inspiratory muscle training was derived from the study of Inzelberg et al. (2005) [14]. In that study, the MIP in the experimental group ($n = 10$) increased from 62±8,2 to 78±7,5, and the control group ($n = 10$) showed no statistically significant difference (51±8.0). Considering a significance level ($\alpha$) of 5% and a power of 0.90, a sample size of ten participants are required. The effect size ($f$) of 0.54 for expiratory muscle training was derived from the study of Sapienza et al. (2011) [15]. In that study, the MEP in the experimental group ($n = 30$) increased from 105.29±28.81 to 133.26±35.53, and the control group ($n = 30$) reduced from 103.65±24.82 to 99.23±27.46. Considering a significance level ($\alpha$) of 5%

and a power of 0.90, fourteen participants per group are required (a total of 28 participants). For analysis of variance, the power is approximately 90% for an effect size of 0.5, considering an $\alpha = 0.05$ and a sample size of 30 individuals [18]. This power was used aiming at a more conservative estimate since larger samples have greater statistical power [18].

Therefore, a sample size of 28 individuals (14 in each group) was defined (largest sample size calculated). Assuming an expected dropout rate of 20%, a total sample size of 34 individuals was set (17 in each group).

## Statistical analysis

All statistical analyzes will be performed by an independent examiner, blinded to the group allocation. Each participant will be assigned to a unique code. All analyses will be performed using SPSS version 25.0 (SPSS Inc., Chicago, IL, USA). The distribution of continuous variables will be checked for normality using the Shapiro-Wilk test. Descriptive statistics will be calculated for all outcomes (means and standard deviations for normally distributed data; medians and interquartile range for continuous nonparametric data; frequency and percentages for categorical data).

The effects of the interventions will be analyzed from the collected data using intention-to-treat. All randomized individuals will be included in the analyses. For dropouts, data from the last available assessment will be used for missed sessions [17, 25]. Two-way ANOVA with repeated measures (2*3) will be used to evaluate the differences between groups, considering the time factor (baseline, post-intervention, and 12-week follow-up), followed by post hoc test selected based on final sample size and in the assumption of equal of variance [18]. If necessary, a similar nonparametric test will be used. The level of significance will be set at 5% [25].

The effect sizes will be calculated to determine the magnitude of the differences between the groups. The differences between the two mean values will be expressed in units of their SD, expressed as Cohen's d, or mean results for the experimental group minus the mean results for the control group, divided by the SD of the control group. Effect sizes between 0.2 and 0.5 will be considered small; between 0.5 and 0.8, medium; and above 0.8, large.

## Ethics

The study obtained ethical approval from the Institutional Research Ethical Committee (CAAE: 53970421.0.0000.5149; number: 5.356.570) of the [information was included in the unmasked version]. All participants will be explained about the objectives and procedures of the study, and will sign a statement of informed consent. The researchers will take all appropriate customary steps to ensure the privacy and confidentiality of the participants, as well as the protection of data. After the publication of the results of this study and the secondary ones, the data underlying the findings will be available without restriction upon request. To ensure confidentiality, data will be blinded to any identifying participant information.

## Discussion

To the best of our knowledge, this will be the first study to investigate the effects of high-intensity respiratory muscle training (combined inspiratory and expiratory muscle training) in individuals with PD. High-intensity respiratory muscle training has the potential to improve inspiratory and expiratory muscle strength in this population. In addition, this can improve inspiratory muscle endurance, dyspnea perception, fatigue, exercise capacity and quality of life. Finally, combined inspiratory and expiratory muscle training using a single device is cheaper and feasible, takes less time and is easily operated, as well as seem to be more effective.

The effect of respiratory muscle training in individuals with PD was already investigated by few studies [12, 17]. A systematic review with meta-analysis examined the effects of non-pharmacological intervention on respiratory impairments in individuals with PD [17]. Due to the small number of published studies and the undetermined effect size, it was not possible to perform a meta-analysis of the effect of respiratory muscle training on inspiratory muscle strength and endurance, and peak cough flow [17]. Therefore, further studies are needed to understand the effects of respiratory muscle training in this population.

In the studies that investigated the effects of respiratory muscle training in individuals with PD, the protocols were very variable. Two randomized clinical trials investigated the effect of inspiratory muscle training on inspiratory strength individuals with PD. Inzelberg et al. 2005 [14] started with an initial training load of 15% of MIP (initial training load), with an increase of 5–10% per week until reaching 60% of the MIP achieved at baseline, at the end of the first month [14]. Then, the training load was adjusted to 60% of the new MIP values, evaluated monthly [14]. The protocol was one daily 30-min sessions, six times a week, for 12 weeks [14]. Reyes et al. 2018 [16] started with an initial training load of 50% of MIP, adjusted every two weeks until reaching 75% of the initial training load. The duration was 5 cycles of 5 breaths per day, sex times a week, for 8 weeks [16].

Two studies investigated the effect of expiratory muscle training in individuals with PD. Sapienza et al. 2011 [15] started with an initial training load of 75% of MEP, 5 sets of 5 repetitions, five times for week, for four weeks [15]. Reyes et al. 2018 [16] started with an initial training load of 50% MIP, adjusted every two weeks until reaching 75% of the initial training load. The duration was 5 cycles of 5 breaths per day, six times for week, for 8 weeks [16].

In other populations, the effects of respiratory muscle training were increased when higher training intensities were applied [21, 22]. Results were better when higher loads (>50% of baseline), session duration (at least 30-min for day), training duration (eight weeks), and progressively increasing loads (adjusted weekly to the new MIP and MEP values) were used [20–22]. Therefore, in the present study the effects of a high-intensity training (60% of MIP and MEP, adjusted weekly to the new values of maximal respiratory pressures, two daily 20-min sessions, seven times a week, for eight weeks) will be examined in individuals with PD.

Individuals with PD commonly present respiratory muscle weakness, both inspiratory and expiratory muscle groups [2, 44]. Inspiratory muscle weakness can lead to impaired coughing, increased perception of dyspnea and reduced exercise capacity [7]. The reduction in expiratory muscle strength may result in decreased subglottic pressure during the compression phase of cough, reduced expiratory airflow, and inadequate airway clearance [4]. This favors the accumulation of secretions and the occurrence of infections, such as pneumonia [4]. Therefore, it is important to train both muscle groups in this population, and this training using a single piece of equipment is more cost-effective, simple and easy to use [20]. However, the effects of combined inspiratory and expiratory muscle training on respiratory muscle strength, inspiratory muscle endurance, peak cough flow, dyspnea, fatigue, exercise capacity, and quality of life have not yet been investigated in this population. In this study, this combination will be performed using the Orygen Dual Valve® device [14, 15, 29].

A recent systematic review showed that, among the instruments used to perform respiratory muscle training, only two allow inspiratory and expiratory muscle training with independent loads [45]. Among them, Orygen Dual Valve® had the advantage of being cost-effective (inexpensive), approximately $70 [45]. In addition, the Orygen Dual Valve® allows proper load adjustments and mouthpiece sealing, is easy to use, portable and allows home-based training [45].

Respiratory muscle training is simple, easy to understand, and can be performed in the community setting, reducing transportation difficulties [20]. Increasing the training volume

(load, duration, or frequency) can enhance the results already reported in the literature [20] and can also improve other important outcomes that have not been yet determined and/or investigated, such as inspiratory muscle endurance, fatigue, exercise capacity, and quality of life. The combined inspiratory and expiratory muscle training using a single device has the advantages of promoting the benefits in both inspiratory and expiratory muscle strengthening, being easier to use, saving time and cost-effective [20]. Clinicians will be able to use the results of this study to implement this training in clinical practice. In addition, researchers may use the results of this study to design other studies.

## Strengthens and weakness of the proposed protocol

This study protocol proposes the use of simple and low-cost measures commonly used in clinical practice to measure outcomes. This intervention will be carried out in the home environment that increases feasibility, accessibility, reduces time, and costs of transport. This increases the feasibility to reproduce their findings in clinical practice. This protocol proposes a randomized clinical trial that is the best study design to investigate the intervention effect. The trial will also have blinded allocation, blinded evaluator, and intention-to-treat analysis. The therapist responsible for carrying out the training has experience in neurological and respiratory rehabilitation and a master's degree in neurological rehabilitation.

This trial also has some limitations. Due to the characteristics of the intervention, participants cannot be completely blinded. Some strategies will be used in an attempt to blind the participants, such as the device will be covered with opaque material to not allow the visualization of the training load. In addition, the intervention will be carried out in the home environment. This requires the commitment and motivation of the participant. Strategies to encourage complete exercise are also planned, such as completing the training diary and weekly visits.

## Supporting information

**S1 File. Complete protocol approved by the ethics committee.**
(PDF)

**S2 File. Complete protocol approved by the ethics committee (in Portuguese).**
(PDF)

**S3 File. SPIRIT checklist.**
(PDF)

## Author Contributions

**Conceptualization:** Sherindan Ayessa Ferreira de Brito, Aline Alvim Scianni, Bruna Mara Franco Silveira, Elem Rodrigues Martins de Oliveira, Maria Eduarda Mateus, Christina Danielli Coelho de Morais Faria.

**Investigation:** Sherindan Ayessa Ferreira de Brito, Aline Alvim Scianni, Bruna Mara Franco Silveira, Elem Rodrigues Martins de Oliveira, Maria Eduarda Mateus, Christina Danielli Coelho de Morais Faria.

**Methodology:** Sherindan Ayessa Ferreira de Brito, Aline Alvim Scianni, Bruna Mara Franco Silveira, Elem Rodrigues Martins de Oliveira, Maria Eduarda Mateus, Christina Danielli Coelho de Morais Faria.

**Project administration:** Sherindan Ayessa Ferreira de Brito, Aline Alvim Scianni, Bruna Mara Franco Silveira, Elem Rodrigues Martins de Oliveira, Maria Eduarda Mateus, Christina Danielli Coelho de Morais Faria.

**Supervision:** Aline Alvim Scianni, Christina Danielli Coelho de Morais Faria.

**Writing – original draft:** Sherindan Ayessa Ferreira de Brito, Aline Alvim Scianni, Bruna Mara Franco Silveira, Elem Rodrigues Martins de Oliveira, Maria Eduarda Mateus, Christina Danielli Coelho de Morais Faria.

**Writing – review & editing:** Sherindan Ayessa Ferreira de Brito, Aline Alvim Scianni, Bruna Mara Franco Silveira, Elem Rodrigues Martins de Oliveira, Maria Eduarda Mateus, Christina Danielli Coelho de Morais Faria.

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
