## [Decision Letter · Decision Letter 0]

8 May 2023

PONE-D-23-05779Effects of high-intensity respiratory muscle training on respiratory muscle strength in individuals with Parkinson's disease: protocol of a randomized clinical trialPLOS ONE

Dear Dr. Faria,

Thank you for submitting your manuscript to PLOS ONE. After careful consideration, we feel that it has merit but does not fully meet PLOS ONE’s publication criteria as it currently stands. Therefore, we invite you to submit a revised version of the manuscript that addresses the points raised during the review process.

We look forward to receiving your revised manuscript.

Kind regards,

Enock Madalitso Chisati, PhD

Academic Editor

PLOS ONE

Journal Requirements:

- https://doi.org/10.1186/s13063-018-2823-0

In your revision ensure you cite all your sources (including your own works), and quote or rephrase any duplicated text outside the methods section. Further consideration is dependent on these concerns being addressed.

"Financial support provided by the Fundação de Amparo à Pesquisa do Estado de Minas Gerais (FAPEMIG), Coordenação de Aperfeiçoamento de Pessoal de Nível Superior (CAPES-Finance code 001), Conselho Nacional de Desenvolvimento Científico e Tecnológico (CNPQ) and Pró-reitoria de Pesquisa da Universidade Federal de Minas Gerais (PRPq/UFMG)."

Reviewers' comments:

Reviewer's Responses to Questions

**Comments to the Author**

1. Does the manuscript provide a valid rationale for the proposed study, with clearly identified and justified research questions?

Reviewer #1: Yes

Reviewer #2: Partly

Reviewer #3: Yes

2. Is the protocol technically sound and planned in a manner that will lead to a meaningful outcome and allow testing the stated hypotheses?

Reviewer #1: Yes

Reviewer #2: Partly

Reviewer #3: Yes

3. Is the methodology feasible and described in sufficient detail to allow the work to be replicable?

Reviewer #1: Yes

Reviewer #2: Yes

Reviewer #3: Yes

4. Have the authors described where all data underlying the findings will be made available when the study is complete?

Reviewer #1: No

Reviewer #2: Yes

Reviewer #3: No

5. Is the manuscript presented in an intelligible fashion and written in standard English?

Reviewer #1: Yes

Reviewer #2: Yes

Reviewer #3: Yes

6. Review Comments to the Author

You may also provide optional suggestions and comments to authors that they might find helpful in planning their study.

Reviewer #1: I recommend researchers state the SPSS version (e.g., IBM SPSS Statistics 28.0). There is need for consistency in the use of the tense (future tense). Clearly state how all data underlying the findings described in the manuscript will be fully available without restriction, with rare exception, at the time of publication.

Reviewer #2: General comment

Thanks for the opportunity to review this manuscript. I have made some comments that I believe will help the authors to improve the protocol.

Specific comments

- In the introduction section. It is not clear by what mechanism reduced inspiratory/expiratory muscle strength and endurance may have an effect in reduced cough efficacy. In particular, the effect of respiratory muscle endurance on cough efficacy is not clear for the reader. Please clarify.

- I suggest to the authors that it is more accurate to state that in case of accumulation of mucus, an effective cough may reduce the risk of aspiration and a respiration infection.

- Dyspnoea in individuals with PD may have multiple causes. It is not clear that dyspnoea is related to inspiratory muscle weakness in individuals with PD. Dyspnoea is described in individuals with inspiratory muscle weakness and cardiometabolic reflex. This is not the case in PD. In addition, individuals with do not necessarily have inspiratory muscle weakness. I suggest being more accurate with the rationale of the study.

- Although previous studies have not reported or calculated effect sizes, as long as these studies provide the mean, standard deviation and sample size, it is possible to calculate the effect sizes. It is not clear what is the point that the authors try to make with this argument.

- In lines 104-105. This statement has no logical connection with the argument of the paragraph. Please note that a topic not yet investigated does not necessarily mean that the topic is relevant for research.

- In lines 125-126. It is not clear how the coordination of inspiratory and expiratory muscles will improve cough effectiveness. Please provide more details

- In the methods section, study design, what do the authors mean by superiority and parallel group trial? Please explain and check if those descriptors are relevant or add relevant information.

- What parameters make the intervention “high intensity”? Is there a number as a criterion for that classification?

- It is not clear why the authors used <80 and <90 cmH20 mip and mep for inclusion criteria. As it is, seems arbitrary.

- Regarding randomization to study groups. I suggest explaining if for example participants will be allocated using a simple randomisation method, the minimization method, bloke randomisation method, etcetera.

- The control group will be using the same training device as the intervention group? Is it possible to set this device at 0 cmH20? Please clarify.

- It is not clear why the authors intent to apply the same protocol to improve strength and endurance. It may be better to use a different protocol for each property.

- There may be no point in wrapping the device for the control group. Training without load will be evident for participants.

- It is certainly possible that not all individuals with PD may be able to start training at 60% of MIP. I suggest to the authors to include familiarisation weeks with the device.

- I suggest to the authors to use a different criterion to increase the training load. This is because the Borg scale may not be sensitive enough to detect the effort of participants.

- Is it important for the authors the number of efforts that participants can make in 20 minutes? It may be that in 20 minutes each participant makes different number of efforts. This may add some extra bias to the protocol. Please take this into consideration.

- I suggest to the authors to revise literature regarding test-retest, reliability and learning effect of mip and mep measurements. There is extensive literature indicating that a single measure of mip and mep underestimates respiratory muscle strength.

- It may be worth including reflex cough flow as an outcome. This is because reflex cough is more ecologically valid than voluntary cough.

- In the sample size calculation please provide the effect sizes used and reported in the studies cited by the authors. The sample size calculation requires more details. What study design the authors are applying the estimated effect sizes? Please note that an effect size for a two-way anova analysis is different than that for two mean comparison test, pre and post comparison, etcetera. Please also indicate the estimated error.

- I suggest providing more details about how the authors will apply the intention to treat principle.

- Please consider the anova design (two groupsXthree time points) plus group and time effect for sample size calculation.

- The Friedman test does not follow after two-way anova for post hoc comparisons. Please modify.

- In the method section of the abstract. Please note that by definition a randomized control trial is prospective. Please avoid redundancies.

Reviewer #3: In this study protocol, a prospective randomized controlled trial is being proposed to investigate the efficacy of high-intensity respiratory muscle training on improving inspiratory and expiratory muscle strength, muscle endurance, peak cough flow, dyspnea, fatigue, exercise capacity, and quality of life in individuals with Parkinson’s disease. Outcomes will be measured at baseline, post-intervention, and at 12-week follow-up.

Minor revisions:

1- Abstract Line 52: Typographical error: consisting instead of consisted.

2- Line 186: Indicate if block randomization will be used. If so, state the block size.

3- Line 305: State the full details of the sample size calculation such that the estimates can be verified. Indicate the effect size.

4- State the descriptive statistics that will be estimated. Typically means and standard deviations are presented for normally distributed data while medians, first and third quartiles are summarized for continuous nonparametric data. Frequency and percentages are summarized for categorical data.

6- Identify the software that will be used to capture the data as well as the software that will be used for the statistical analysis.

7. PLOS authors have the option to publish the peer review history of their article (what does this mean?). If published, this will include your full peer review and any attached files.

Reviewer #1: **Yes: **BHEKUZULU KHUMALO

Reviewer #2: No

Reviewer #3: No

---

## [Author Response · Author response to Decision Letter 0]

28 Jun 2023

Review Comments to the Author

We would like to thank you for the positive considerations regarding our manuscript. You provided excellent suggestions that helped to improve our work. All your comments, suggestions, and corrections were carefully taken into consideration and point-by-point responses are included below.

Reviewer #1: I recommend researchers state the SPSS version (e.g., IBM SPSS Statistics 28.0). There is need for consistency in the use of the tense (future tense). Clearly state how all data underlying the findings described in the manuscript will be fully available without restriction, with rare exception, at the time of publication.

As suggested, this information has been added to the manuscript, as follows:

Line 336-337: “All analyzes will be performed using SPSS version 25.0 (SPSS Inc., Chicago, IL, USA).”

We would like to apologize for these mistakes. The manuscript was revised to ensure the correct use of the tense (future tense), as follows:

Line 225-228: “Borg score of dyspnea and effort [27] will be considered for adjusting training intensity, and scores from 4 to 6 will be targeted. Borg score will be asked after one set of three minutes, and if Borg score is <4 or >6 the training load will be adjusted to higher or lower than 60% of MIP and MEP [28].”

As suggested, this information was included in the manuscript text, as follows:

Line 362-365: "After the publication of the results of this study and the secondary ones, the data underlying the findings will be available without restriction upon request. To ensure confidentiality, data will be blinded to any identifying participant information."

Reviewer #2: General comment

Thanks for the opportunity to review this manuscript. I have made some comments that I believe will help the authors to improve the protocol.

Specific comments

- In the introduction section. It is not clear by what mechanism reduced inspiratory/expiratory muscle strength and endurance may have an effect in reduced cough efficacy. In particular, the effect of respiratory muscle endurance on cough efficacy is not clear for the reader. Please clarify.

The relationship is between respiratory muscle weakness and peak cough flow. The text of the manuscript was revised to clarify this point. Cough is the main mechanism of protection of the airways, a process that occurs through a high-velocity airflow. This airflow is generated by the contraction of the expiratory muscles while the glottis is closed. Reduced expiratory muscle strength results in decreased subglottic pressure during the compression phase, reduced initial cough expiratory airflow rate, decreased cough peak expiratory airflow, reduced plateau phase expiratory airflow, and inadequate clearance of material from the airway. 

We are sorry for this mistake. As suggested, the text of the manuscript was revised to fix this information, as follows:

Line 76-81: “Reduced respiratory muscle strength is associated with an impaired cough efficacy [4]. Cough is the main mechanism of protection of the airways, a process that occurs through a high-velocity airflow. This airflow is generated by the contraction of the expiratory muscles while the glottis is closed [4,5]. Expiratory muscle weakness results in reduced subglottic pressure during the compression phase, decreased expiratory airflow, and inadequate clearance of material from the airway [4,5].”

- I suggest to the authors that it is more accurate to state that in case of accumulation of mucus, an effective cough may reduce the risk of aspiration and a respiration infection.

Pulmonary aspiration syndromes result from aspiration of foreign material into the lungs, which can cause aspiration pneumonia.1 This does not occur only when there is an accumulation of mucus. Aspiration pneumonia can occur due to the aspiration of various foreign bodies, such as saliva and food. Microaspiration is recognized as the main pathogenic mechanism in pneumonia where particulate material and microorganisms can enter the upper airways and then reach the lower airways and respiratory tract.2 Macroaspiration, refers to the aspiration of a large volume of oropharyngeal or upper gastrointestinal content passing through the trachea and larynx into the lungs.2 This aspiration triggers an inflammation and immune response in the lungs. Coughing is the main protective mechanism of the airways, which eliminates foreign bodies through a high-velocity airflow.3 Cough dysfunction has been demonstrated in asymptomatic PD subjects, which may contribute to aspiration and increased risk of pneumonia.4 In addition, individuals with PD commonly have dysphagia and sialorrhea, which, together with cough dysfunction, significantly increase the risk of aspiration.4

1 Rodriguez AE, Restrepo MI. New perspectives in aspiration community acquired Pneumonia. Expert Rev Clin Pharmacol. 2019 Oct;12(10):991-1002. 

2 Niederman MS, Cilloniz C. Aspiration pneumonia. Rev Esp Quimioter. 2022 Apr;35 Suppl 1(Suppl 1):73-77 

3 Lee KK, Davenport PW, Smith JA, Irwin RS, McGarvey L, Mazzone SB, Birring SS; CHEST Expert Cough Panel. Global Physiology and Pathophysiology of Cough: Part 1: Cough Phenomenology - CHEST Guideline and Expert Panel Report. Chest. 2021 Jan;159(1):282-293. 

4 D'Arrigo A, Floro S, Bartesaghi F, Casellato C, Sferrazza Papa GF, Centanni S, Priori A, Bocci T. Respiratory dysfunction in Parkinson's disease: a narrative review. ERJ Open Res. 2020 Oct 5;6(4):00165-2020. doi: 10.1183/23120541.00165-2020. PMID: 33043046; PMCID: PMC7533305.

As suggested, the text of the manuscript was revised to clarify this information, as follows:

Line 79-84: “Expiratory muscle weakness results in reduced subglottic pressure during the compression phase, decreased expiratory airflow, and inadequate clearance of material from the airway [4, 5]. This favors the aspiration of foreign bodies and accumulation of mucus, and consequently, the occurrence of respiratory infections, such as pneumonia [5]. Individuals with PD have a significant reduction of cough efficacy when compared to controls [5].”

- Dyspnoea in individuals with PD may have multiple causes. It is not clear that dyspnoea is related to inspiratory muscle weakness in individuals with PD. Dyspnoea is described in individuals with inspiratory muscle weakness and cardiometabolic reflex. 

This is not the case in PD. In addition, individuals do not necessarily have inspiratory muscle weakness. I suggest being more accurate with the rationale of the study.

Dyspnea is a disabling non-motor symptom that occurs in a significant proportion of PD patients.1 Several mechanisms may underpin dyspnea in the PD patient.1 These are multifactorial and involve a complex interplay between the effects of the disease on the respiratory and cardiovascular system, sensory and perceptual changes, psychological factors, medication effects, and other associated co-morbidities in the individual patient.1 It is likely that during periods of increased respiratory demand such as exercise, there is an inability to meet the higher demand by the respiratory muscles, causing dyspnea.1 In addition, inspiratory weakness may also contribute to nocturnal hypoventilation and dyspnea.1 Studies have already shown that individuals with Parkinson's disease commonly have respiratory muscle weakness.2,3

The effects of respiratory muscle training on dyspnea have already been demonstrated in other populations, such as individuals with heart failure4 and stroke.5 In individuals with PD, there is a growing body of evidence to support the benefits of inspiratory/expiratory muscle training.1 A previous study compared the effects of a 12-week inspiratory muscle training program with sham training in two groups of 10 PD patients and demonstrated significant improvements in inspiratory muscle strength and endurance in the intervention group.6 Furthermore, the perception of dyspnea improved, suggesting that inspiratory muscle training may also have modulatory effects on somatosensory pathways.6 However, in this study, only inspiratory muscle training was performed.

1 Vijayan S, Singh B, Ghosh S, Stell R, Mastaglia FL. Dyspnea in Parkinson's disease: an approach to diagnosis and management. Expert Rev Neurother. 2020 Jun;20(6):619-626. 

2 Santos RBD, Fraga AS, Coriolano MDGWS, Tiburtino BF, Lins OG, Esteves ACF, et al. Respiratory muscle strength and lung function in the stages of Parkinson's disease. J Bras Pneumol. 2019;45(6):e20180148. 

3 Weiner P, Inzelberg R, Davidovich A, Nisipeanu P, Magadle R, Berar-Yanay N, et al. Respiratory muscle performance and the Perception of dyspnea in Parkinson's disease. Can J Neurol Sci. 2002;29(1):68-72. doi: 10.1017/s031716710000175x.

4 Marco E, Ramírez-Sarmiento AL, Coloma A, Sartor M, Comin-Colet J, Vila J, Enjuanes C, Bruguera J, Escalada F, Gea J, Orozco-Levi M. High-intensity vs. sham inspiratory muscle training in patients with chronic heart failure: a prospective randomized trial. Eur J Heart Fail. 2013 Aug;15(8):892-901. 

5 Parreiras de Menezes KK, Nascimento LR, Ada L, Avelino PR, Polese JC, Mota Alvarenga MT, et al. High-Intensity Respiratory Muscle Training Improves Strength and Dyspnea Poststroke: A Double-Blind Randomized Trial. Arch Phys Med Rehabil. 2019;100(2):205-212. 

6 Inzelberg R, Peleg N, Nisipeanu P, Magadle R, Carasso RL, Weiner P. Inspiratory muscle training and the perception of dyspnea in Parkinson's disease. Can J Neurol Sci. 2005;32(2):213-7. doi: 10.1017/s0317167100003991. 

As suggested, the text of the manuscript was revised, as follows:

Line 90-97: “Dyspnea is multifactorial and arises from the interaction of several factors, such as the effects of the disease on the respiratory and cardiovascular systems, sensory and perceptual changes, and other associated comorbidities [7]. This may result from a mismatch between central respiratory motor activity and income afferent information from receptors in the airways, chest wall structures, and respiratory muscles [7,10]. The inability of the respiratory muscles to meet the increased respiratory demand, as occurs during activity performance, can cause dyspnea [7].”

- Although previous studies have not reported or calculated effect sizes, as long as these studies provide the mean, standard deviation and sample size, it is possible to calculate the effect sizes. It is not clear what is the point that the authors try to make with this argument.

According to our knowledge, only one study investigated the effects of inspiratory muscle training for the investigated outcomes (inspiratory muscle endurance and quality of life). This study did not show the effect size and the data provided did not allow its calculation. Data from the control group after training were not showed (mean and standard deviation).

As suggested, this information was detailed in the manuscript, as follows:

Line 104-107: “To our knowledge, only one study investigated the effects of inspiratory muscle training for the following outcomes (inspiratory muscle endurance and quality of life). However, this study did not present the effect size and the data provided did not allow its calculation [12, 14, 17].”

- In lines 104-105. This statement has no logical connection with the argument of the paragraph. Please note that a topic not yet investigated does not necessarily mean that the topic is relevant for research.

Respiratory muscle weakness has already been associated with the occurrence of fatigue and reduced exercise capacity in individuals with PD. The cited paragraph presents a literature review on the effects of respiratory muscle training in this population. However, the effects of respiratory muscle training on fatigue and exercise capacity in this population have not yet been investigated. It is important to present this information, as these outcomes will be evaluated in this study.

As suggested, this paragraph was revised to clarify this connection, as follows:

Line 100-114: “Respiratory muscle training has been used to increase inspiratory and expiratory muscle strength in people with PD [12,13]. It was found only three randomized controlled trials that have investigated the effects of respiratory muscle training in individuals with PD [14-16]. These studies showed low methodological quality when evaluated by the PEDro scale [12,14-16]. To our knowledge, only one study investigated the effects of inspiratory muscle training for the following outcomes (inspiratory muscle endurance, dyspnea and quality of life). However, this study did not show the effect size and the data provided did not allow its calculation [12, 14, 17]. The effect size provides information about the magnitude of the effects of an intervention, being an estimate of the size of the difference between groups [18]. Interventions that have larger effect sizes are more likely to produce significant outcomes for patients. Therefore, it is essential to determine the effect size of the interventions that will be used in clinical practice [18]. Finally, although respiratory muscle weakness is associated with the fatigue and reduced exercise capacity in individuals with PD [2,11], the effects of respiratory muscle training on these outcomes have not yet been investigated.”

- In lines 125-126. It is not clear how the coordination of inspiratory and expiratory muscles will improve cough effectiveness. Please provide more details.

Initiation of a cough most commonly results in a large inspiration followed by multiple expulsive events during the decrease in expired volume The magnitude of the inspiratory volume is also proportional to the number of cough reaccelerations. Active expiratory muscle contraction and elastic recoil of the thoracic system against a closed glottis result in a rapidly increasing subglottic pressure. Therefore, for an effective cough, the coordination between inspiratory and expiratory muscles is necessary.

As suggested, this information was detailed in the manuscript as follows:

Line 135-137: “In addition, initiation of a cough usually results in a large inspiration followed by expulsive events, in which the expiratory muscles contract. Therefore, so for an effective cough, the coordination between inspiratory and expiratory muscles is necessary [4].”

- In the methods section, study design, what do the authors mean by superiority and parallel group trial? Please explain and check if those descriptors are relevant or add relevant information.

Superiority: When the aim of the randomized controlled trial (RCT) is to show that one treatment is superior to another, a statistical test is employed and the trial (test) is called a superiority trial.1

Parallel group trial: A parallel group design is an experimental study design in which each subject is randomized to one of two or more distinct treatment/intervention groups. Those who are assigned to the same treatment are referred to as a treatment group.

While the treatments that these groups receive differ, all groups are treated as equally as possible in all other regards, and they complete the same procedures during the study.2

According to the SPIRIT checklist, in the topic of study design, the following should be presented: a succinct description that conveys the topic (study population, interventions), acronym (if any), and basic study design—including the method of intervention allocation (eg, parallel group randomised trial; single-group trial)—will facilitate retrieval from literature or internet searches and rapid judgment of relevance. It can also be helpful to include the trial framework (eg, superiority, non-inferiority), study objective, or primary outcome.3

1 Lesaffre E. Superiority, equivalence, and non-inferiority trials. Bull NYU Hosp Jt Dis. 2008;66(2):150-4. PMID: 18537788.

2 Turner, J.R. (2013). Parallel Group Design. In: Gellman, M.D., Turner, J.R. (eds) Encyclopedia of Behavioral Medicine. Springer, New York, NY. 

3 Chan AW, Tetzlaff JM, Gøtzsche PC, Altman DG, Mann H, Berlin JA, Dickersin K, Hróbjartsson A, Schulz KF, Parulekar WR, Krleza-Jeric K, Laupacis A, Moher D. SPIRIT 2013 explanation and elaboration: guidance for protocols of clinical trials. BMJ. 2013 Jan 8;346:e7586. doi: 10.1136/bmj.e7586. PMID: 23303884; PMCID: PMC3541470.

- What parameters make the intervention “high intensity”? Is there a number as a criterion for that classification?

Load is the main criterion described in the literature to define training intensity, with a load >50% of maximum respiratory pressures considered high-intensity training.

As suggested, this data was clarified in the text, as follows:

Line 401-404: “Results were better when higher loads (>50% of baseline), session duration (at least 30-min for day), training duration (eight weeks), and progressively increasing loads (adjusted weekly to the new MIP and MEP values) were used [20-22].”

- It is not clear why the authors used <80 and <90 cmH20 mip and mep for inclusion criteria. As it is, seems arbitrary.

In the ATS/ERS statement on respiratory muscle testing, a MIP of less than -80 cmH2O was proposed as a practical threshold to exclude clinically important inspiratory muscle weakness.1 

For individuals with neurological conditions, it has been suggested that MIP less than <80 cmH2O, or MEP greater than <90 cmH2O excludes significant muscle weakness.2 Therefore, these values were assumed in this study.

1 Laveneziana P, Albuquerque A, Aliverti A, Babb T, Barreiro E, Dres M, Dubé BP, Fauroux B, Gea J, Guenette JA, Hudson AL, Kabitz HJ, Laghi F, Langer D, Luo YM, Neder JA, O'Donnell D, Polkey MI, Rabinovich RA, Rossi A, Series F, Similowski T, Spengler CM, Vogiatzis I, Verges S. ERS statement on respiratory muscle testing at rest and during exercise. Eur Respir J. 2019 Jun 13;53(6):1801214. 

2 Farrero E, Antón A, Egea CJ, Almaraz MJ, Masa JF, Utrabo I, Calle M, Verea H, Servera E, Jara L, Barrot E, Casolivé V; Sociedad Española de Neumología y Cirugía Torácica (SEPAR). Guidelines for the management of respiratory complications in patients with neuromuscular disease. Sociedad Española de Neumología y Cirugía Torácica (SEPAR). Arch Bronconeumol. 2013 Jul;49(7):306-13. English, Spanish. doi: 10.1016/j.arbres.2012.12.003. Epub 2013 Feb 12. PMID: 23410743.

As suggested, the information was clarified in the manuscript, as follows: 

Line 180-182: “MIP <80cmH2O or MEP <90cmH2O (suggested values to eliminate significant muscle weakness) [25].”

- Regarding randomization to study groups. I suggest explaining if for example participants will be allocated using a simple randomisation method, the minimization method, bloke randomisation method, etcetera.

As suggested, this information was explained in the text, as follows:

Line 200-202: “A trained research assistant will generate the randomization sequence prior to the commencement of the study. Individuals will be randomly allocated through blinded, computer-generated randomization in six-participant blocks.”

- The control group will be using the same training device as the intervention group? Is it possible to set this device at 0 cmH20? Please clarify.

Both groups will use the same instrument. Yes, it is possible to set this device at 0cmH20. The Orygen Dual Valve® (Forumed S.L., Girona, Spain) device provides inspiratory workloads from 0cmH2O to 70cmH2O and expiratory workloads from 0cmH2O to 80cmH2O.

As suggested, this information was clarified in the text, as follows:

Line 214-217: “The Orygen Dual Valve® (Forumed S.L., Girona, Spain) device will be used to carry out the intervention, which provides inspiratory workloads from 0cmH2O to 70cmH2O and expiratory workloads from 0cmH2O to 80cmH2O.”

- It is not clear why the authors intent to apply the same protocol to improve strength and endurance. It may be better to use a different protocol for each property.

Respiratory muscle strength and endurance are different concepts and, for some time, it was believed that strength improvement is achieved with high intensity and lower frequency, and endurance training with low intensity and higher frequency. The currently recommended respiratory muscle training allows gains in the two dimensions mentioned, considering that stronger muscles are also more efficient when contracting at low percentages of the maximum when compared to weaker muscles. Studies that performed the same protocol in other populations found significant improvements in both strength and endurance.1,2

1 Parreiras de Menezes KK, Nascimento LR, Ada L, Avelino PR, Polese JC, Mota Alvarenga MT, et al. High-Intensity Respiratory Muscle Training Improves Strength and Dyspnea Poststroke: A Double-Blind Randomized Trial. Arch Phys Med Rehabil. 2019;100(2):205-212. 

2 Hoffman M, Augusto VM, Eduardo DS, Silveira BMF, Lemos MD, Parreira VF. Inspiratory muscle training reduces dyspnea during activities of daily living and improves inspiratory muscle function and quality of life in patients with advanced lung disease. Physiother Theory Pract. 2021 Aug;37(8):895-905. 

- There may be no point in wrapping the device for the control group. Training without load will be evident for participants.

Blinding is a challenge in this type of study. The Orygen Dual Valve® (Forumed S.L., Girona, Spain) provides a load of 0-70cmH2O for MIP and 0-80cmH2O for MEP, with load increments every 10cmH2O. The minimum load of 10cmH2O may already have training effects for weaker individuals, which would represent an important bias for the study. Therefore, as in previous studies, a load of 0cmH2O was selected.1,2,3 

The device will be covered with opaque material in both groups in an attempt to blind the participants. In a previous study with a similar protocol, about 58% and 47% of the experimental and control groups, respectively, thought that they were in the experimental group, with most of the remaining participants being unsure.1

1 Parreiras de Menezes KK, Nascimento LR, Ada L, Avelino PR, Polese JC, Mota Alvarenga MT, et al. High-Intensity Respiratory Muscle Training Improves Strength and Dyspnea Poststroke: A Double-Blind Randomized Trial. Arch Phys Med Rehabil. 2019;100(2):205-212.

2 Sapienza C, Troche M, Pitts T, Davenport P. Respiratory strength training: concept and intervention outcomes. Semin Speech Lang. 2011;32(1):21-30. doi: 10.1055/s-0031-1271972.

3 Troche MS, Okun MS, Rosenbek JC et al. Aspiration and swallowing in Parkinson disease and rehabilitation withEMST: a randomized trial. Neurology. 2010;75(21):1912-9.

- It is certainly possible that not all individuals with PD may be able to start training at 60% of MIP. I suggest to the authors to include familiarisation weeks with the device.

Two criteria will be used to guarantee the training intensity (60% of the maximum respiratory pressures and the Borg score). Therefore, if the Borg score is <4 or >6 the training load will be adjusted to as close as possible to 60% of MIP and MEP, and these data will be recorded.

- I suggest to the authors to use a different criterion to increase the training load. This is because the Borg scale may not be sensitive enough to detect the effort of participants.

The main criterion used to increase the workload is the value of maximum respiratory pressures evaluated weekly. The Borg score will be an additional criterion used to guarantee the training intensity. Once a week, the trained researcher will visit their homes to perform the necessary procedures to guarantee the intensity of the training. On this visit, MIP and MEP will be evaluated and the training load will be progressed to ensure that 60% of the new pressure values are maintained. Borg score will be asked after one set of three minutes, and if the Borg score is <4 or >6 the training load will be adjusted to as close as possible to 60% of MIP and MEP.

As suggested, the text of the manuscript was revised to clarify this information, as follows:

Line 218-225: "Participants in the experimental group will perform high-intensity inspiratory and expiratory muscle training (60% of MIP and MEP). The initial training load for each participant will be set at 60% of his/her maximal baseline MIP and MEP for both inspiratory and expiratory strength training, respectively. Once a week, the trained researcher will visit their homes to perform the necessary procedures to guarantee the intensity of the training. On this visit, MIP and MEP will be evaluated and the training load will be progressed to ensure that 60% of the new pressure values are maintained [16,29].”

- Is it important for the authors the number of efforts that participants can make in 20 minutes? It may be that in 20 minutes each participant makes different number of efforts. This may add some extra bias to the protocol. Please take this into consideration.

The Orygen Dual Valve® device (Forumed S.L., Girona, Spain) does not provide the number of repetitions performed, like the POWERbreathe®. To record the number of repetitions, subjects would have to count the number of breaths while performing the training. This is home-based training without direct therapist supervision. Therefore, this count could make training unfeasible and/or favor the occurrence of errors. Furthermore, both groups (experimental and control) are equally susceptible to the occurrence of these differences.

Several other randomized controlled trials used protocols similar to that of the present study.1,2,3 There is no evidence of superiority between protocols in which the duration is defined by the number of repetitions compared to those defined by time.

1 Parreiras de Menezes KK, Nascimento LR, Ada L, Avelino PR, Polese JC, Mota Alvarenga MT, et al. High-Intensity Respiratory Muscle Training Improves Strength and Dyspnea Poststroke: A Double-Blind Randomized Trial. Arch Phys Med Rehabil. 2019;100(2):205-212.

2 Inzelberg R, Peleg N, Nisipeanu P, Magadle R, Carasso RL, Weiner P. Inspiratory muscle training and the perception of dyspnea in Parkinson's disease. Can J Neurol Sci. 2005;32(2):213-7. doi: 10.1017/s0317167100003991. 

- I suggest to the authors to revise literature regarding test-retest, reliability and learning effect of mip and mep measurements. There is extensive literature indicating that a single measure of mip and mep underestimates respiratory muscle strength.

The main reference used to design the respiratory measurements evaluation protocol was the ERS Statement on Respiratory Muscle Testing at Rest and during Exercise.1 According to the ERS Statement, learning effects need to be acknowledged and sufficient baseline trials (at least 5 maneuvers) have to be performed.1 Therefore, in this study, the tests will be carried out following the recommendation: 

Line 255-258: "A minimum of five measurements will be performed, with a one-minute rest between them. A difference of less than 10% must be achieved among the three best measurements and the last measurement of the test cannot be the greatest."

1 Laveneziana P, Albuquerque A, Aliverti A, Babb T, Barreiro E, Dres M, et al. ERS statement on respiratory muscle testing at rest and during exercise. Eur Respir J. 2019;53(6):1801214. doi: 10.1183/13993003.01214-2018.

- It may be worth including reflex cough flow as an outcome. This is because reflex cough is more ecologically valid than voluntary cough.

Thanks for the suggestion. However, the assessment and intervention performed in this study were designed to be easily replicated in clinical practice. The evaluation of the peak flow of voluntary cough is more feasible to be performed in clinical contexts. It provides valid and reliable measurements of peak cough flow. Therefore, this measure meets the objectives of this study well.

- In the sample size calculation please provide the effect sizes used and reported in the studies cited by the authors. The sample size calculation requires more details. What study design the authors are applying the estimated effect sizes? Please note that an effect size for a two-way anova analysis is different than that for two mean comparison test, pre and post comparison, etcetera. 

The sample size calculation was performed in the G*Power Software® (version 3.1) considering the primary outcome measures (inspiratory and expiratory muscle strength). The sample size calculation was performed using data from two randomized clinical trials that measured the effects of respiratory muscle training on respiratory muscle strength in individuals with PD (Inzelberg et al., 2005 and Sapienza et al., 2011). The study design applied was the ANOVA repeated measures, between factors.

As suggested, the sample size text was revised and more details were added, as follows:

Line 316-333: “The sample size was estimated using the G*Power Software® (version3.1) and considering data provided by previous randomized controlled trials that investigated the effects of respiratory muscle training on respiratory muscle strength in individuals with PD [14,15]. 

The effect size (f) of 1.00 for inspiratory muscle training was derived from the study of Inzelberg et al. (2005) [14]. In that study, the MIP in the experimental group (n=10) increased from 62±8,2 to 78±7,5, and the control group (n=10) showed no statistically significant difference (51±8.0). Considering a significance level (α) of 5% and a power of 0.90, a sample size of ten participants are required. The effect size (f) of 0.54 for expiratory muscle training was derived from the study of Sapienza et al. (2011) [15]. In that study, the MEP in the experimental group (n=30) increased from 105.29±28.81 to 133.26±35.53, and the control group (n=30) reduced from 103.65±24.82 to 99.23±27.46. Considering a significance level (α) of 5% and a power of 0.90, fourteen participants per group are required (a total of 28 participants). Therefore, a sample size of 28 individuals (14 in each group) was defined (largest sample size calculated). Assuming an expected dropout rate of 20%, a total sample size of 34 individuals was set (17 in each group).”

- I suggest providing more details about how the authors will apply the intention to treat principle.

As suggested, this information was detailed in the manuscript, as follows:

Line 342-345: The effects of the interventions will be analyzed from the collected data using intention-to-treat. All randomized individuals will be included in the analyses. For dropouts, data from the last available assessment will be used for missed sessions [17,25].

- Please consider the anova design (two groupsXthree time points) plus group and time effect for sample size calculation.

The sample size was calculated using group and time effect. A partial n2 of 0.06 (moderate effect) was used. Considering a 0.05 and a power of 80%, a sample size of 28 individuals was determined. Therefore, this sample size is planned for the present study.

- The Friedman test does not follow after two-way anova for post hoc comparisons. Please modify.

Analysis of variance are based on several assumptions about the nature of data, such as normal distribution of data.1 In general, ANOVA is robust to violations of these assumptions, so that it can be used with confidence in most research situations.1 In this study, two-way ANOVA with repeated measures (2*3) will be used to evaluate differences between groups, considering time factor, followed by post-hoc tests. If ANOVA could not be used with confidence for any variable, a similar nonparametric test will be used. In this case, a statistic professional will perform the analysis. As suggested, the text of statistical analyzes were revised, as follows:

Line 345-348: “Two-way ANOVA with repeated measures (2*3) will be used to evaluate the differences between groups, considering the time factor (baseline, post-intervention, and 12-week follow-up), followed by post-hoc tests. If necessary, a similar nonparametric test will be used.”

1 Portney LG. Foundations of clinical research: applications to evidence-based practice. FA Davis. 2020.

- In the method section of the abstract. Please note that by definition a randomized control trial is prospective. Please avoid redundancies.

As suggested, the text was revised, as follows:

Line 46: “Methods: A randomized controlled trial, concealed allocation, blinded assessments…”

 

Reviewer #3: In this study protocol, a prospective randomized controlled trial is being proposed to investigate the efficacy of high-intensity respiratory muscle training on improving inspiratory and expiratory muscle strength, muscle endurance, peak cough flow, dyspnea, fatigue, exercise capacity, and quality of life in individuals with Parkinson’s disease. Outcomes will be measured at baseline, post-intervention, and at 12-week follow-up.

Minor revisions:

Abstract Line 52: Typographical error: consisting instead of consisted.

We are sorry for this mistake. As suggested, the error has been fixed, as follows:

Line 51-53: “Individuals will perform a home-based intervention, with indirect home supervision, consisting of two daily 20-min sessions (morning and afternoon), seven times a week, during eight weeks.”

Line 186: Indicate if block randomization will be used. If so, state the block size.

As suggested, this information was explained in the text, as follows:

Line 200-202: “A trained research assistant will generate the randomization sequence prior to the commencement of the study. Individuals will be randomly allocated through blinded, computer-generated randomization in six-participant blocks.”

Line 305: State the full details of the sample size calculation such that the estimates can be verified. Indicate the effect size.

The sample size calculation was performed in the G*Power Software® (version 3.1) considering the primary outcome measures (inspiratory and expiratory muscle strength). The sample size calculation was performed using data from two randomized clinical trials that measured the effects of respiratory muscle training on respiratory muscle strength in individuals with PD (Inzelberg et al., 2005 and Sapienza et al., 2011). The study design applied was the ANOVA repeated measures, between factors. As suggested, the sample size text was revised and more details were added, as follows:

Line 317-333: “The sample size was estimated using the G*Power Software® (version3.1) and considering data provided by previous randomized controlled trials that investigated the effects of respiratory muscle training on respiratory muscle strength in individuals with PD [14,15]. 

The effect size (f) of 1.00 for inspiratory muscle training was derived from the study of Inzelberg et al. (2005) [14]. In that study, the MIP in the experimental group (n=10) increased from 62±8,2 to 78±7,5, and the control group (n=10) showed no statistically significant difference (51±8.0). Considering a significance level (α) of 5% and a power of 0.90, a sample size of ten participants are required. The effect size (f) of 0.54 for expiratory muscle training was derived from the study of Sapienza et al. (2011) [15]. In that study, the MEP in the experimental group (n=30) increased from 105.29±28.81 to 133.26±35.53, and the control group (n=30) reduced from 103.65±24.82 to 99.23±27.46. Considering a significance level (α) of 5% and a power of 0.90, fourteen participants per group are required (a total of 28 participants). Therefore, a sample size of 28 individuals (14 in each group) was defined (largest sample size calculated). Assuming an expected dropout rate of 20%, a total sample size of 34 individuals was set (17 in each group).”

State the descriptive statistics that will be estimated. Typically means and standard deviations are presented for normally distributed data while medians, first and third quartiles are summarized for continuous nonparametric data. Frequency and percentages are summarized for categorical data.

As suggested, this information was included in the manuscript text, as follows:

Line 338-341: “Descriptive statistics will be calculated for all outcomes (means and standard deviations for normally distributed data; medians and interquartile range for continuous nonparametric data; frequency and percentages for categorical data).”

Identify the software that will be used to capture the data as well as the software that will be used for the statistical analysis.

As suggested, this information has been added to the manuscript, as shown below:

Line 336-337: “All analyzes will be performed using SPSS version 25.0 (SPSS Inc., Chicago, IL, USA).”

---

## [Decision Letter · Decision Letter 1]

4 Aug 2023

PONE-D-23-05779R1Effects of high-intensity respiratory muscle training on respiratory muscle strength in individuals with Parkinson's disease: protocol of a randomized clinical trialPLOS ONE

Dear Dr. Faria,

Thank you for submitting your manuscript to PLOS ONE. After careful consideration, we feel that it has merit but does not fully meet PLOS ONE’s publication criteria as it currently stands. Therefore, we invite you to submit a revised version of the manuscript that addresses the points raised during the review process.

We look forward to receiving your revised manuscript.

Kind regards,

Enock Madalitso Chisati, PhD

Academic Editor

PLOS ONE

Journal Requirements:

Reviewers' comments:

Reviewer's Responses to Questions

**Comments to the Author**

1. Does the manuscript provide a valid rationale for the proposed study, with clearly identified and justified research questions?

Reviewer #2: Yes

Reviewer #3: Yes

2. Is the protocol technically sound and planned in a manner that will lead to a meaningful outcome and allow testing the stated hypotheses?

Reviewer #2: Yes

Reviewer #3: Yes

3. Is the methodology feasible and described in sufficient detail to allow the work to be replicable?

Reviewer #2: Yes

Reviewer #3: Yes

4. Have the authors described where all data underlying the findings will be made available when the study is complete?

Reviewer #2: Yes

Reviewer #3: Yes

5. Is the manuscript presented in an intelligible fashion and written in standard English?

Reviewer #2: Yes

Reviewer #3: Yes

6. Review Comments to the Author

You may also provide optional suggestions and comments to authors that they might find helpful in planning their study.

Reviewer #2: the authors have addressed all my comments. I have no further comments. My recommendation is to accept the manuscript

Reviewer #3: Minor revisions:

1- Line 321: Indicate the statistical testing method(s) which attain 90% power.

2- Line 347: State the statistical method that will be used for the post-hoc tests.

3- Line 345: Clarify if the primary test of interest is the interaction effect of group by time.

4- Line 336: The following statement is unclear. Does this imply that the distribution of continuous variables will be checked for normality? If so, state the method(s) that will used. "The normality of data distribution will be for all continuous numeric variables."

Note: Line numbers refer to those in the tracked changes version of revision 1.

7. PLOS authors have the option to publish the peer review history of their article (what does this mean?). If published, this will include your full peer review and any attached files.

Reviewer #2: **Yes: **Alvaro Reyes Ponce

Reviewer #3: No

---

## [Author Response · Author response to Decision Letter 1]

14 Aug 2023

We would like to thank you for the positive considerations regarding our manuscript. You provided excellent suggestions that helped to improve our work. All your comments, suggestions, and corrections were carefully taken into consideration and point-by-point responses are included below.

Reviewer #3: Minor revisions:

1- Line 321: Indicate the statistical testing method(s) which attain 90% power.

Power is the probability that a test will lead to rejection of the null hypothesis, or the probability of reaching statistical significance (Portney & Watkins, 2015). Power can be analyzed to estimate sample size or determine the probability that a Type II error has been made when a study results in a non-significant finding (Portney & Watkins, 2015). Researchers can estimate sample size by determining a significance level, effect size, and power. This is called an a priori power analysis (Portney & Watkins, 2015), which was used in the present protocol.

 Two-way ANOVA is the statistical test planned to be used to analyze the data of the randomized controlled trial. The power can be determined for each of the main effects and interaction effects of the two-way ANOVA (Portney & Watkins, 2015). According to the tables provided by Portney & Watkins (2015) for analysis of variance, the power is approximately 90% for an effect size of 0.5, considering an α = 0.05 and a sample size of 30 individuals (Portney & Watkins, 2015). This power was used aiming at a more conservative estimate since larger samples have greater statistical power (Portney & Watkins, 2015).

As suggested, this information has been added to the manuscript, as follows:

Line 230-233: “For analysis of variance, the power is approximately 90% for an effect size of 0.5, considering an α = 0.05 and a sample size of 30 individuals [18]. This power was used aiming at a more conservative estimate since larger samples have greater statistical power [18].”

2- Line 347: State the statistical method that will be used for the post-hoc tests.

In most post hoc analyses, the specific comparisons of interest are decided after the analysis of variance has been completed and are considered unplanned comparisons (Portney, 2020). The choice of the post-hoc tests must be done considering some characteristics such as equal or unequal sample size and the assumption (or not) of equal variances. Fisher’s Least Significant Difference (LSD), Tukey’s Honestly Significant Difference (HSD) or Bonferroni are some of the statistical methods that can be used for the post-hoc tests (Portney, 2020). However, only after the analysis of variance has been completed, it will be possible to correctly select the post-hoc test. 

As suggested, the manuscript text was revised, as follows:

Line 248-251: “Two-way ANOVA with repeated measures (2*3) will be used to evaluate the differences between groups, considering the time factor (baseline, post-intervention, and 12-week follow-up), followed by post hoc test selected based on final sample size and in the assumption of equal of variance [18].”

3- Line 345: Clarify if the primary test of interest is the interaction effect of group by time.

Between-group statistical comparison considering the interaction effect of group by time (baseline, post-intervention, and 12-week follow-up) is the primary test of interest. 

This information is described in text, as follows: 

Line 248-251: “Two-way ANOVA with repeated measures (2*3) will be used to evaluate the differences between groups, considering the time factor (baseline, post-intervention, and 12-week follow-up), followed by post hoc test selected based on final sample size and in the assumption of equal of variance [18].”

4- Line 336: The following statement is unclear. Does this imply that the distribution of continuous variables will be checked for normality? If so, state the method(s) that will used. "The normality of data distribution will be for all continuous numeric variables."

As suggested, the text of the manuscript was revised, as follows: 

Line 337-338: “The distribution of continuous variables will be checked for normality using the Shapiro-Wilk test.”

---

## [Decision Letter · Decision Letter 2]

22 Aug 2023

Effects of high-intensity respiratory muscle training on respiratory muscle strength in individuals with Parkinson's disease: protocol of a randomized clinical trial

PONE-D-23-05779R2

Dear Dr. Faria,

We’re pleased to inform you that your manuscript has been judged scientifically suitable for publication and will be formally accepted for publication once it meets all outstanding technical requirements.

Kind regards,

Enock Madalitso Chisati, PhD

Academic Editor

PLOS ONE

Additional Editor Comments (optional):

Reviewers' comments:

Reviewer's Responses to Questions

**Comments to the Author**

1. Does the manuscript provide a valid rationale for the proposed study, with clearly identified and justified research questions?

Reviewer #3: Yes

2. Is the protocol technically sound and planned in a manner that will lead to a meaningful outcome and allow testing the stated hypotheses?

Reviewer #3: Yes

3. Is the methodology feasible and described in sufficient detail to allow the work to be replicable?

Reviewer #3: Yes

4. Have the authors described where all data underlying the findings will be made available when the study is complete?

Reviewer #3: No

5. Is the manuscript presented in an intelligible fashion and written in standard English?

Reviewer #3: Yes

6. Review Comments to the Author

You may also provide optional suggestions and comments to authors that they might find helpful in planning their study.

Reviewer #3: All comments have been adequately addressed.

7. PLOS authors have the option to publish the peer review history of their article (what does this mean?). If published, this will include your full peer review and any attached files.

Reviewer #3: No

---

## [Editor Report · Acceptance letter]

29 Aug 2023

PONE-D-23-05779R2 

Effects of high-intensity respiratory muscle training on respiratory muscle strength in individuals with Parkinson's disease: protocol of a randomized clinical trial 

Dear Dr. Faria:

I'm pleased to inform you that your manuscript has been deemed suitable for publication in PLOS ONE. Congratulations! Your manuscript is now with our production department. 

Kind regards, 

on behalf of

Dr. Enock Madalitso Chisati 

Academic Editor

PLOS ONE